# Effect of CO_2_ Elevation on Tomato Gas Exchange, Root Morphology and Water Use Efficiency under Two N-Fertigation Levels

**DOI:** 10.3390/plants13172373

**Published:** 2024-08-26

**Authors:** Manyi Zhang, Wentong Zhao, Chunshuo Liu, Changtong Xu, Guiyu Wei, Bingjing Cui, Jingxiang Hou, Heng Wan, Yiting Chen, Jiarui Zhang, Zhenhua Wei

**Affiliations:** 1Key Laboratory of Agricultural Soil and Water Engineering in Arid and Semiarid Areas of Ministry of Education, Northwest A&F University, Yangling 712100, China; zhangmanyinwafu@126.com (M.Z.); 15225760909@163.com (W.Z.); lcs20@nwafu.edu.cn (C.L.); 2023055918@nwafu.edu.cn (C.X.); 2019060353@nwafu.edu.cn (G.W.); cbj123@nwafu.edu.cn (B.C.); hengwan@nwafu.edu.cn (H.W.); zjr9813@nwafu.edu.cn (J.Z.); 2Department of Plant and Environmental Sciences, Faculty of Science, University of Copenhagen, Højbakkegaard Allé 13, DK-2630 Taastrup, Denmark; yiting@plen.ku.dk; 3State Key Laboratory of Efficient Utilization of Agricultural Water Resources, Beijing 100083, China; jxhrl818@163.com; 4Soil Physics and Land Management Group, Wageningen University, P.O. Box 47, 6700 AA Wageningen, The Netherlands

**Keywords:** elevated CO_2_, N-fertigation, gas exchange, root morphology, water use efficiency, tomato

## Abstract

Atmospheric elevated CO_2_ concentration (*e*[CO_2_]) decreases plant nitrogen (N) concentration while increasing water use efficiency (WUE), fertigation increases crop nutrition and WUE in crop; yet the interactive effects of *e*[CO_2_] coupled with two N-fertigation levels during deficit irrigation on plant gas exchange, root morphology and WUE remain largely elusive. The objective of this study was to explore the physiological and growth responses of ambient [CO_2_] (*a*[CO_2_], 400 ppm) and *e*[CO_2_] (800 ppm) tomato plant exposed to two N-fertigation regimes: (1) full irrigation during N-fertigation (FIN); (2) deficit irrigation during N-fertigation (DIN) under two N fertilizer levels (reduced N (N1, 0.5 g pot^−1^) and adequate N (N2, 1.0 g pot^−1^). The results indicated that *e*[CO_2_] associated with DIN regime induced the lower N2 plant water use (7.28 L plant^−1^), maintained leaf water potential (−5.07 MPa) and hydraulic conductivity (0.49 mol m^−2^ s^−1^ MPa^−1^), greater tomato growth in terms of leaf area (7152.75 cm^2^), specific leaf area (223.61 cm^2^ g^−1^), stem and total dry matter (19.54 g and 55.48 g). Specific root length and specific root surface area were increased under N1 fertilization, and root tissue density was promoted in both *e*[CO_2_] and DIN environments. Moreover, a smaller and denser leaf stomata (4.96 µm^2^ and 5.37 mm^−2^) of N1 plant was obtained at *e*[CO_2_] integrated with DIN strategy. Meanwhile, this combination would simultaneously reduce stomatal conductance (0.13 mol m^−2^ s^−1^) and transpiration rate (1.91 mmol m^−2^ s^−1^), enhance leaf ABA concentration (133.05 ng g^−1^ FW), contributing to an improvement in WUE from stomatal to whole-plant scale under each N level, especially for applying N1 fertilization (125.95 µmol mol^−1^, 8.41 µmol mmol^−1^ and 7.15 g L^−1^). These findings provide valuable information to optimize water and nitrogen fertilizer management and improve plant water use efficiency, responding to the potential resource-limited and CO_2_-enriched scenario.

## 1. Introduction

CO_2_ concentration in atmosphere ([CO_2_]) has successively elevated after industrial revolution and would largely influence crop water use and production around the growth areas [1]. Moreover, reasonable water and nitrogen (N) application were two important agricultural management strategies for limited resource utilization in arid regions [2]. Thereby, efficient N fertigation during deficit irrigation was crucial for improving resource efficiency and sustainable agricultural development responding to future elevated [CO_2_] (*e*[CO_2_]) environment.

Previous study has revealed that *e*[CO_2_] was able to increase the plant photosynthetic rate (P_n_), but reduce stomatal conductance (g_s_) due to the enhanced ratio of CO_2_ to O_2_ in the chloroplast and guard cell depolarization, respectively [3]. These further lowered the transpiration rate (T_r_) and hydraulic conductivity, but was beneficial to optimize stomatal morphology, improve leaf water potential and plant water use efficiency (WUE) [4]. The recent study has also presented that endogenous abscisic acid (ABA) level played an important part in mediating plant g_s_ response to water stress at *e*[CO_2_] [5]. In addition, *e*[CO_2_] was studied to promote root development and vary root architecture system, especially the stimulated lateral root growth to facilitate the water use and nutrient acquisition [6,7]. However, N concentration was observed to be declined in *e*[CO_2_] plant, mostly owing to the N dilution effect on enlarged leaf area and dry matter, and decreased mass flow in soil nutrient on account of the lower leaf T_r_ at *e*[CO_2_] [8]. This would have a partly adverse effect on plant physiology and WUE under *e*[CO_2_] circumstance.

It was well established that deficit irrigation (DI) as a water-saving method induced temporal soil water dynamics, transported the root-derived ABA signaling in drying soil to leaf to reduce g_s_ and water use, modulate plant expansion rate and water status, thereby enhancing WUE [9]. Furthermore, compared to conventional irrigation and fertilization supply, fertigation technique provided irrigation water and nutrients concurrently into rooting soil [10], minimizing fertilizer loss beyond the root growth zone and decreasing soil compaction to maintain a suitable soil water and nutrient condition, hence achieving precise irrigation and fertilizer application [11]. This uniform, timed and frequent fertigation could directly optimize root morphology, maximize soil root activity and nutrient bioavailability, and synergistically promote water and nutrient use efficiency of plant [12].

N was the necessary component for plant physiology and growth requirement in relation to the primary constituent of chlorophyll, amino acids and proteins, etc. [13]. Generally, an increased N supply/leaf N concentration was largely connected with higher photosynthetic ability, leading to an increased promoted P_n_ than g_s_ and T_r_, consequently improving leaf area, plant dry biomass and WUE [14,15]. Previous evidence has noted that N deficiency induced the higher stomatal sensitivity to plant ABA, indicating that ABA was participated in the modulation of N signal transduction, lateral root expansion and plant N accumulation [16]. Additionally, N availability could change the ontogeny and distribution of root system [17]. Insufficient N supply was studied to stimulate primary and lateral root elongation [18], whereas there was earlier study reported that adequate N concentration enlarged the entire root system and accordingly promoted N uptake and the photosynthetic process [19]. Hereby, these controversial explorations in root architecture and plant growth under different N fertilizer levels would need to be further investigated.

As for fertigation practice, the recent report has observed that medium N supply during fertigation showed the similar production and WUE as adequate N supply [20]. Whereas [21] observed that greater N fertilizer quantity under fertigation enhanced yield and WUE, it was plant stomatal aperture, not N status, mostly regulating WUE during fertigation, and coupled moderate water stress and sufficient N fertilizer in fertigation was recommended for improving WUE of plant [22]. Hereby, applying DI associated with N-fertigation at different N supply levels may be adopted as an original irrigation and N management strategy for continuable resource utilization and agricultural development in *e*[CO_2_] scenario. Nevertheless, the interactive effect of *e*[CO_2_] in combination with N-fertigation regime, especially under varied N application levels on plant gas exchange, root growth and water utilization was poorly documented, and it was of great interest to fully explore the beneficial impact of N-fertigation technique on *e*[CO_2_] plant WUE.

Tomato was widely cultivated and produced in recent years as the most popular and second consumed vegetable crop around the world [23]. The enhanced popularity was mainly ascribed to its bright color, favorite flavor, abundant mineral nutrients and attractive taste, being beneficial for our daily diet and health. Additionally, tomato was the phytochemical source of iron, vitamin C and antioxidants, such as phenolic compounds and lycopene in relation to anticancer property and reduction in angiocardiopathy risk [24]. Therefore, the purpose of this study was to investigate the effect of N-fertigation during deficit irrigation on response of leaf gas exchange, water relation, root morphology and tomato plant WUE subjected to two N supply levels (reduced N (N1, 0.5 g pot^−1^) and adequate N fertilizer (N2, 1.0 g pot^−1^) at ambient [CO_2_] (*a*[CO_2_], 400 ppm) and *e*[CO_2_] (800 ppm) condition. Explicitly, leaf gas exchange, water relations, stomatal morphology, ABA concentration, root morphological traits, plant water use and dry matter were examined. It was hypothesized that implementation of N-fertigation during deficit irrigation in *e*[CO_2_] plant could conjointly optimize stomatal and root morphology, further enhancing WUE at different scales under both N fertilizer levels, particularly for reduced N supply.

## 2. Materials and Methods

### 2.1. Experimental Setup

This climate-controlled experiment was conducted at Northwest A&F University, Yangling, Shaanxi, China. The tomato (Jingpeng 14-8) seedlings were applied in 4.0 L pots (height of 19.5 cm, lower diameter of 16 cm and upper diameter of 19.5 cm) with 1.0 kg peat soil (Substrate, Pindstrup Mosebrug A/S, Ryomgård, Denmark). The water-holding capacity (WHC) of the pot was measured according to that described in [3]. The two climate-controlled chambers were maintained at 23/16 ± 2 °C day/night temperature and 60% relative air humidity, 16 h photoperiod was controlled by automatic curtain and 500 µmol m^−2^ s^−1^ photosynthetic flux density was supplied by LED plus high-pressure sodium lamps.

### 2.2. Treatments

The climate-controlled chambers were achieved with two CO_2_ concentrations, namely ambient CO_2_ concentration (*a*[CO_2_]) with 400 ppm and elevated CO_2_ concentration (*e*[CO_2_]) with 800 ppm. [CO_2_] concentration was maintained by CO_2_ emission from a gas cylinder, monitored by an automatic control system in each chamber and distributed uniformly via internal ventilation.

Tomato seeds were sown on 21st of April 2022. Two weeks after sowing, one four-leaf stage of healthy seedlings was transplanted to each 4.0 L pot in two chambers and constantly irrigated to well-watered at 95% WHC levels (i.e., 3.0 kg of pot weight) in three weeks. A 2 cm layer of perlite was covered on the soil surface to reduce evaporation. Thereafter, tomato plants were exposed to two N-fertigation regimes: full irrigation during N-fertigation (FIN) and deficit irrigation during N-fertigation (DIN) where the pot was irrigated daily to 95% and 70% of WHC at 17:00 h, respectively. All pots were weighed using the analytical balance and manually irrigated to reach the FIN and DIN treatment requirement.

Tomato plants were separated to two N fertilizer levels, i.e., N1 fertilization using 0.5 g N pot^−1^ and N2 fertilization using 1.0 g N pot^−1^ as implemented with CO(NH_2_)_2_ during the whole experimental period. In addition, 0.5 g P and 0.63 g K were implemented with KH_2_PO_4_ to each tomato plant. Then, 25% N fertilizer, all P and K fertilizers were fully dissolved as the base fertilizers and evenly watered into the pot soil. The 75% remaining N fertilizer (N1: 0.375 g N, N2: 0.75 g N) was applied during the N-fertigation period, namely N fertilization was transported by irrigation water on 5, 10, 16, 21, 26 days under N-fertigation regime with a total of five times. The N-fertigation treatments lasted six weeks. This experiment included eight treatments in total, and four replicated plants were implemented for each N-fertigation and N fertilizer level treatment.

### 2.3. Measurements

#### 2.3.1. Leaf Gas Exchange

P_n_, g_s_ and T_r_ in leaf were determined on the fully expended upper leaves in the morning of the 35th day after fertigation treatment with the LiCor-6800 Portable Photosynthesis System (LI-Cor, Lincoln, NE, USA). The measured condition was set at 23 °C leaf temperature, 1200 mol m^−2^ s^−1^ photosynthetic flux density, 400 ppm and 800 ppm [CO_2_] concentration, respectively. Leaf intrinsic and instantaneous WUE (WUE_i_ and WUE_n_) during N-fertigation treatment was estimated as the ratio of P_n_ to g_s_ and P_n_ to T_r_, respectively.

#### 2.3.2. Leaf Water Relation and ABA

The same fresh leaf for measuring gas exchange was invested to determine leaf water potential (Ψ_l_) with the pressure chamber (SEC3005, Soil Moisture Equipment, Santa Barbara, CA, USA). Leaf hydraulic conductivity (K_l_) was estimated as |Ψ_l_| divided by T_r_. The Ψ_l_ measured leaf was removed and frozen at −80 °C for determining ABA concentration in leaf ([ABA]_leaf_) with the ELISA approach according to [25].

#### 2.3.3. Stomatal Aperture and Density

On the 40th day after the fertigation experiment, the material of silicone impression (Zhermack S.p.A., Badia Polesine, Italy) was used to obtain a leaf epidermal impression from the abaxial and adaxial surface of fully expanded upper leaf. The imprints were transferred to the microscope using clear cellophane tape and observed with the light microscope (BA210 Digital, MOTIC, Xiamen, China). The Image J2 software (1.6.0-24; Wayne Rasband, National Institutes of Health, Kensington, MD, USA) was used to reveal images on the computer screen and measure stomatal aperture (SA), containing stomatal pore aperture width (W_a_) and aperture length (L_a_). Four images (each with a 320 × 240 μm calibrated size) of each imprint were taken for counting stomata (namely stomatal density (SD)).

#### 2.3.4. Root Morphology

The fresh root samples were cleanly washed and thereafter collected for scanning with WinRHIZO Pro (Modified Epson Expression 12000XL, Regent Instruments Inc., Québec, QC, Canada) at the resolution of 400 dots per inch on a 20 × 25 cm transparent tray. Thereafter, the standard edition WinRHIZO software was applied to analyze root images for obtaining the root morphological traits: root length (RL), root diameter (RD), root surface area (RSA) and root volume (RV). In addition, the specific root length (SRL), specific root surface area (SRSA), root length density (RLD) and root tissue density (RTD) were estimated as the ratio of RL to root dry matter, RSA to root dry matter, RL to volume of soil and root dry matter to RV, respectively.

#### 2.3.5. Leaf Area and Plant Water Use Efficiency

At the end of the experimental period, leaf area (LA) was obtained by the Leaf Area Meter (3100, LI-COR Inc., Lincoln, NE, USA). Each plant sample was separated into leaf, stem and root samples for drying at 70 °C to the constant dry matter weight. Specific leaf area (SLA) was estimated as the ratio of LA to leaf dry matter. Total dry matter increment (TDM) was estimated as the summarized dry matter increment of leaf, stem and root during the fertigation experimental period. Plant water use (PWU) was the water use amounts in plant during the fertigation period. Plant water use efficiency (WUE_p_) was estimated as PWU divided by TDM.

#### 2.3.6. Statistics Analysis

The SPSS (IBM 22 software, New York, NY, USA) was used to statistically assess three-way analysis of variance (ANOVA) about the effect of CO_2_ concentrations ([CO_2_]), N-fertigation regimes ([FN]) and N fertilizer levels ([N]) and their interaction on P_n_, g_s_, T_r_, Ψ_l_, K_l_, SA, SD, [ABA]_leaf_, WUE_i_, WUE_n_, leaf area, SLA, root morphology variables, each tissue and total dry matter, PWU and WUE_p_. Error bars denoted the standard error of four replications. *, ** and *** denoted the notable levels at *p* < 0.05, *p* < 0.01 and *p* < 0.001, respectively.

Pearson’s relation among the variables and principle component analysis (PCA) were presented by Origin Pro 2022 (OriginLab Inc., Northampton, MA, USA).

## 3. Results

### 3.1. Gas Exchange and WUE at Leaf Level

*e*[CO_2_], FIN and N2 (19.37 µmol m^−2^ s^−1^, 18.18 µmol m^−2^ s^−1^ and 16.95 µmol m^−2^ s^−1^) had notably increased 54.52%, 32.49% and 13.41% P_n_ related to *a*[CO_2_], DIN and N1 (12.53 µmol m^−2^ s^−1^, 13.72 µmol m^−2^ s^−1^ and 14.95 µmol m^−2^ s^−1^), respectively (Figure 1a, Table 1). Both g_s_ and T_r_ were notably decreased 29.46% and 28.60%, 27.16% and 25.90% under DIN and N1 (0.146 mol m^−2^ s^−1^ and 0.147 mol m^−2^ s^−1^, 2.12 mmol m^−2^ s^−1^ and 2.14 mmol m^−2^ s^−1^) compared with those under FIN and N2 (0.207 mol m^−2^ s^−1^ and 0.206 mol m^−2^ s^−1^, 2.91 mmol m^−2^ s^−1^ and 2.89 mmol m^−2^ s^−1^), respectively (Figure 1b,c, Table 1).

[ABA]_leaf_ at *e*[CO_2_], DIN and N2 (136.17 ng g^−1^ FW, 134.11 ng g^−1^ FW and 133.53 ng g^−1^ FW) was obviously 8.18%, 4.83% and 3.90% greater than that at *a*[CO_2_], FIN and N1 (125.88 ng g^−1^ FW, 127.93 ng g^−1^ FW and 128.52 ng g^−1^ FW), respectively (Figure 1d, Table 1). *e*[CO_2_] and N1 (117.92 µmol mol^−1^ and 103.65 µmol mol^−1^, 8.25 µmol mmol^−1^ and 7.07 µmol mmol^−1^) had obvious 61.57% and 18.79%, 63.76% and 13.72% greater WUE_i_ and WUE_n_ related to *a*[CO_2_] and N2 (72.99 µmol mol^−1^ and 87.25 µmol mol^−1^, 5.04 µmol mmol^−1^ and 6.22 µmol mmol^−1^), respectively (Figure 1e,f, Table 1).

### 3.2. Plant Water Relation and Stomatal Morphology

The Ψ_l_ under N2 (−5.31 MPa) was 16.25% higher than that under N1 (−6.34 MPa) (Figure 2a, Table 1). K_l_ under FIN and N2 (0.54 mol m^−2^ s^−1^ MPa^−1^ and 0.56 mol m^−2^ s^−1^ MPa^−1^) was 45.42% and 57.12% greater than that grown under DIN and N1 (0.37 mol m^−2^ s^−1^ MPa^−1^ and 0.35 mol m^−2^ s^−1^ MPa^−1^), respectively (Figure 2b, Table 1).

SA under N2 (5.99 µm^2^) was 25.84% larger than that under N1 (4.76 µm^2^) (Figure 2c, Table 1). *e*[CO_2_] (5.50 mm^−2^) notably increased 13.64% SD related to *a*[CO_2_] (4.84 mm^−2^) (Figure 2d, Table 1).

### 3.3. Root Morphology

There was no significant effect on tomato RL and RLD. In comparison with N2 (2294.18 cm^2^ and 19.59 cm^3^, 59.54 m g^−1^ and 632.35 cm^2^ g^−1^) plant, N1 (2071.85 cm^2^ and 15.99 cm^3^, 71.33 m g^−1^ and 686.44 cm^2^ g^−1^) plant decreased 9.69% and 18.38% RSA and RV, increased 19.80% and 8.55% SRL and SRSA. *e*[CO_2_] and N1 (319.15 µm and 312.16 µm) had 5.95% and 9.87% smaller RD compared to *a*[CO_2_] and N2 treatment (339.34 µm and 346.34 µm), respectively. *e*[CO_2_] and DIN (198.12 mg cm^−3^ and 193.60 mg cm^−3^) increased 10.83% and 5.63% RTD related to *a*[CO_2_] and FIN treatment (178.75 mg cm^−3^ and 183.28 mg cm^−3^), respectively (Table 2).

### 3.4. Leaf Area and Specific Leaf Area

*e*[CO_2_] and N2 (5270.50 cm^2^ and 5793.88 cm^2^) tomato obviously enhanced 27.52% and 60.51% LA related to *a*[CO_2_] and N1 treatment (4133.06 cm^2^ and 3609.69 cm^2^), respectively (Figure 3a, Table 3). In comparison with *a*[CO_2_] (171.74 cm^2^ g^−1^), *e*[CO_2_] (216.75 cm^2^ g^−1^) improved 26.21% SLA across FN and N treatments (Figure 3b, Table 3).

### 3.5. Dry Matter and WUE at Plant Level

At both CO_2_ environments, DIN and N2 (25.05 g and 30.39 g) increased 5.81% and 65.82% tomato leaf dry matter (LDM) as compared to FIN and N1 treatment (23.67 g and 18.33 g), respectively (Figure 4a, Table 3). Root dry matter (RDM) was only 20.20% higher under N2 (3.63 g) than that under N1 treatment (3.02 g) (Figure 4c, Table 3). Both stem and total dry matter (SDM and TDM) were obviously enhanced 16.51%, 5.53% and 13.28%, 7.40%, 5.61% and 40.25% at *e*[CO_2_], DIN and N2 (17.38 g, 16.58 g and 17.16 g, 45.40 g, 45.03 g and 51.18 g) than those at *a*[CO_2_], FIN and N1 treatment (14.92 g, 15.72 g and 15.14 g, 42.27 g, 42.64 g and 36.49 g), respectively (Figure 4b,d, Table 3).

*e*[CO_2_] and DIN (5.88 L plant^−1^ and 6.63 g L^−1^) decreased 30.17% PWU, but increased 43.51% WUE_p_ related to *a*[CO_2_] and FIN (8.42 L plant^−1^ and 4.62 g L^−1^). Compared to N2 (8.44 L plant^−1^, 5.28 g L^−1^, 5.74 g L^−1^), N1 treatment (5.55 L plant^−1^, 5.16 g L^−1^, 6.56 g L^−1^) decreased 34.24% PWU, decreased 2.27% WUE_p_ at *a*[CO_2_], but increased 14.29% WUE_p_ at *e*[CO_2_] (Figure 4e,f, Table 3).

### 3.6. Principal Component Analysis (PCA) of Tomato Physiology, Growth and WUE

The PCA plots of tomato A_n_, g_s_, T_r_, Ψ_l_, K_l_, SD, SA, [ABA]_leaf_, WUE_i_, WUE_n_, leaf area, SLA, root morphology variables, each tissue and total dry matter, PWU and WUE_p_ (26 parameters in total) are depicted in Figure 5. PC1, PC2 under *a*[CO_2_] and *e*[CO_2_] could explain 43.1%, 21.1% and 40.3%, 23.6% of these 26 parameters, respectively. The clusters of N fertilizer levels were more dispersed than N-fertigation regimes in two [CO_2_] concentrations.

For *a*[CO_2_] (Figure 5a), N1 treatment mainly converged to the left part of the PCA plot in accordance with the vector direction of SD, Ψ_l_, WUE_i_, WUE_n_, SRL and RTD. N2 treatment mainly converged to the right part of the PCA plot in accordance with the vector direction of leaf physiology and plant growth variables. Considering N-fertigation regime, the clustering of [N2, FIN] treatment mostly distributed in the upper right side, indicating greater plant dry matter parameters. Additionally, the clustering of [N2, DIN] treatment mostly distributed in the lower right side, indicating greater leaf gas exchange rates.

For *e*[CO_2_] (Figure 5b), N1 treatment mainly converged to the left part of the PCA plot in accordance with the vector direction of SD, Ψ_l_, WUE_i_, SLA, SRL, SRSA and WUE_p_. N2 treatment mainly converged to the right part of the PCA plot in accordance with the vector direction of leaf physiology and plant growth variables. Considering the N-fertigation regime, the clustering of [N1, FIN] treatment mostly distributed in the upper left side, indicating higher SD, SLA, SRL and SRSA. The clustering of [N1, DIN] treatment mostly distributed in the lower left side, indicating greater Ψ_l_, WUE_i_ and WUE_p_. In addition, the clustering of [N2, FIN] treatment mostly distributed in the upper right side, indicating higher leaf gas exchange rates and root morphology. The clustering of [N2, DIN] treatment mostly distributed in the lower right side, indicating greater SA, [ABA]_leaf_, LA and plant tissue dry matter.

## 4. Discussion

It was well known that *e*[CO_2_] plant commonly had a greater photosynthetic capacity, lower g_s_ and T_r_ as compared to *a*[CO_2_] plant, leading to a promoted WUE at leaf level [3]. Similarly, the finding from this study revealed that *e*[CO_2_] notably enhanced tomato leaf P_n_, WUE_i_ and WUE_n_ related to *a*[CO_2_] environment (Figure 1a,e,f, Table 1), being mainly ascribed to the increased intercellular CO_2_ concentration and stimulated carboxylation reaction through Rubisco activity at *e*[CO_2_] [26]. Meanwhile, *e*[CO_2_] could decrease leaf g_s_ and T_r_ under each N2 and FIN treatment (Figure 1b,c, Table 1) due to the guard cell depolarization [27]. Previous study has presented that ABA was involved in the stomatal modulation under *e*[CO_2_] environment and the close sensitivity of stomata to *e*[CO_2_] was reduced following lower ABA concentration [28]. However, in this study, *e*[CO_2_] had the higher endogenous [ABA]_leaf_ level related to *a*[CO_2_] (Figure 1d, Table 1), which was in line with the recent studies reported that endogenous ABA level played a vital part in regulating plant g_s_ response to water stress at *e*[CO_2_], although this effect was species-dependent [29].

In the present study, leaf P_n_, g_s_ and T_r_ under DIN was declined as compared with that under the FIN regime, being mostly attributed to the stomatal limitation on leaf photosynthetic activity induced by the enhancement in [ABA]_leaf_ and decline in g_s_ under deficit irrigation (Figure 1a–d, Table 1), in agreement with the result of earlier findings [30], leading to the maintaining of both WUE at leaf level (WUE_i_ and WUE_n_) between the FIN and DIN regimes (Figure 1e,f, Table 1). This might be ascribed to the less reduction in leaf g_s_ and T_r_ under DIN treatment as a result of more efficient water and N use as applying N fertigation during deficit irrigation [22]. Additionally, reduced N fertilizer could decrease the leaf photosynthetic process and have a negative effect on stomatal aperture and plant growth [15]. Consistent with this, here compared with adequate N supply (N2), reduced N supply (N1) treatment decreased leaf P_n_, g_s_ and T_r_ across [CO_2_] and [FN] environments, whereas N1 supply increased both leaf WUE (Figure 1e,f, Table 1), owing to a relative larger reduction in g_s_, T_r_ than P_n_ caused by limited N fertilizer. Accordingly, *e*[CO_2_] combined with the DIN regime could decrease g_s_ and T_r_, increase [ABA]_leaf_ and further improve both WUE at leaf scale under N supply, particularly under the N1 fertilizer level. This was also confirmed by the PCA plots (Figure 5b) in this study.

Earlier evidence has obtained that *e*[CO_2_] may be beneficial to retain water and sustain/enhance leaf water relation resulted from the lower g_s_ [31]. Here, in comparison with *a*[CO_2_] plant, *e*[CO_2_] plant maintained Ψ_l_, while decreased K_l_ under the FIN regime and maintained K_l_ under the DIN regime (Figure 2a,b, Table 1). Additionally, the proper lowered soil water content caused by DI treatment could less decline or sustain leaf water status in a variety of plants [32]. Whereas, it was reported that insufficient N supply would accelerate the soluble carbohydrate accumulation to decrease leaf Ψ_l_ and promote osmotic adjustment [13]. Likewise, here it was found that Ψ_l_ was maintained in DIN plant, but reduced in N1 plant. As for K_l_, it was lower under DIN and N1 supply in relation to FIN and N2 supply, respectively, except for DIN plant under N2 together with *e*[CO_2_] environment (Figure 2a,b, Table 1). This was mostly due to the greater decline in T_r_ relative to Ψ_l_ among N-fertigation and N fertilizer levels. Briefly, *e*[CO_2_] coupled with the DIN regime could sustain both leaf Ψ_l_ and K_l_ under N2 supply.

Plant stomata played a crucial role in mediating carbon and water flux via air–leaf interface [3]. It was widely accepted that *e*[CO_2_] preferred to reduce SA in the short term, alter SD and/or SA in the long term, thereby optimizing stomatal morphology to highly mediate the gas exchange process in leaf [33]. In this study, irrespective of N-fertigation and N fertilizer levels, although *e*[CO_2_] had no effect on SA, a notable greater SD was noted in *e*[CO_2_] tomato plant (Figure 2c,d, Table 1), being consistent with our recent finding of maize plant. Earlier evidence has noted that moderate water stress increased SD, but severe water stress decreased SD [34]. Contrast to this study, here there was similar SA and SD between FIN and DIN treatment (Figure 2c,d, Table 1), meaning that N-fertigation under the DI regime may facilitate the water and N uptake to retain stomatal morphology related to the FI regime. Moreover, lower SA was reported to be associated with the insufficient N fertilization [35]. In consensus with this, here N1 supply possessed smaller SA than that under N2 supply (Figure 2c,d, Table 1). Thereby, under N1 fertilizer supply, a smaller and denser leaf stomata was obtained at *e*[CO_2_] coupled with the DIN regime. This might induce a stronger leaf stomatal modulation over gas exchange and further obviously improve WUE of tomato plant.

Accumulated studies have revealed that *e*[CO_2_] could stimulate the leaf area (LA) elongation rate [8], whereas decline SLA in plant [36]. In the current study, *e*[CO_2_] plant had greater LA and SLA than those at *a*[CO_2_] concentration (Figure 3, Table 3). The larger SLA resulted from the greater increase in leaf elongation rate than carbon assimilation rate responding to *e*[CO_2_], probably due to N-fertigation treatment at *e*[CO_2_] facilitating N absorption to promote leaf area and growth in this study. In addition, adequate N supply increased LA as the leaf development was sensitive to the N fertilizer level [37], but SLA was not different under both N levels and N-fertigation regimes. The LA was also similar between the FIN and DIN regimes (Figure 3, Table 3), implying that N-fertigation could simultaneously enhance the soil water and N absorption to sustain leaf Rubisco activity and chlorophyll concentration, and maintaining photosynthetic capacity and leaf assimilates accumulation under both FIN and DIN regimes. Hence, regardless of irrigation regime, *e*[CO_2_] together with adequate N-fertigation had the growth-mediated ability to increase leaf area and SLA in tomato plant.

Root system was well known to largely participate in regulating plant water and nutrient acquisition response to the varied environmental conditions [38]. Generally, the root development at *e*[CO_2_] was stimulated with affected morphological plasticity. Here, *e*[CO_2_] possessed the thinner RD, equivalent RL, RSA, RV, SRL and SRSA, higher RTD related to *a*[CO_2_] tomato (Table 2), indicating that *e*[CO_2_] could change root architecture and enhance root activity to efficiently obtain the water and nutrient resources, and decrease in RD was considered as an adapting root function under *e*[CO_2_] environment [39]. Moreover, root morphological traits were no different between FIN and DIN regimes, except for the greater RTD in DIN treatment (Table 2). The similar root development variables were advised to the identical capacity of water and nutrient uptake directly in uniform soil under the N-fertigation regime [7]. The increased RTD might be ascribed to the fact that *e*[CO_2_] and DIN plants could have a more rapid water and nutrient exploration via the relative higher allocation of root dry matter per unit in root volume as compared with *a*[CO_2_] and FIN plants [40], leading to the greater tomato WUE. In the present study, reduced N (N1) supply declined RSA and RV, but enhanced SRL and SRSA (Table 2). The stimulated root occurrence was connected with the accelerated synthesis of auxin and inhibited synthesis of ethylene in plant [41], in contrast to the earlier study reported that low N fertilization promoted the entire root growth and enlarged the space of the root system [18]. Whereas, sufficient N supply was also suggested to induce primary and lateral root elongation, modulate root architecture and accordingly improve the water and N uptake ability of tomato plant. This root discrepancy of N levels merited further investigation.

Numerous evidence demonstrated that *e*[CO_2_] environment had the prominent C fertilizer effect on plant biomass accumulation [30]. Consistent with this, although LDM and RDM of *e*[CO_2_] were not notably increased as regards *a*[CO_2_] from this study, *e*[CO_2_] plant possessed both significantly higher SDM and TDM than *a*[CO_2_] plant (Figure 4a–d, Table 3), owing to the enhanced photosynthetic assimilation rate in *e*[CO_2_] leaf [11]. It was highly recognized that plant biomass under DI commonly showed no remarkable decline as compared to that under the FI technique [42]. Most interestingly, here DIN tomato had similar RDM, greater LDM, SDM and TDM related to FIN tomato (Figure 4a–d, Table 3), meaning that N fertigation at DI regime was considered to promote the plant photosynthetic process resulted from the efficient N uptake under each N fertilizer level, particularly under N1 supply.

Insufficient N fertilization could restrain the synthesis of chlorophyll and carbonic anhydrase activity, resulting in negative controlled stomatal aperture and plant growth [43]. Thus, the plant dry matter and PWU in N1 tomato were lower than those in N2 tomato (Figure 4a–e, Table 3). Additionally, PWU was also reduced under both *e*[CO_2_] and DIN treatments as a result of the relative decline in leaf g_s_ and T_r_ at *e*[CO_2_] and significant decrease under the DIN regime (Figure 4e, Table 3). The improvement in WUE_p_ of *e*[CO_2_] and DIN tomato was closely relevant to the greater TDM and lower PWU as compared to that of *a*[CO_2_] and FIN tomato, respectively (Figure 4f, Table 3), in line with the previous finding reported in tomato study [26]. For N fertilization level, the N1 plant WUE_p_ was obviously higher than the N2 plant due to the larger decrease degree in PWU relative to TMD of N1 supply (Figure 4f, Table 3). Therefore, consistent with the results revealed in Figure 5b, *e*[CO_2_] associated with the DIN regime could increase SDM and TDM, decrease PWU of N2 plant, and consequently enhance plant WUE under both N levels, especially for N1 fertilization.

## 5. Conclusions

Collectively, *e*[CO_2_] together with the DIN regime could reduce PWU, sustain leaf Ψ_l_ and K_l_, further promote leaf area, SLA and SDM as well as TDM in N2 tomato. The SRL and SRSA were enlarged by N1 fertilizer, and greater RTD was obtained under both *e*[CO_2_] and DIN conditions. In addition, *e*[CO_2_] coupled with DIN strategy induced the lower SA, but higher SD of N1 plant, synergistically decreased g_s_ and T_r_, increased [ABA]_leaf_, leading to the improved WUE at stomatal, leaf and plant scales using two N fertilizer levels, particularly under N1 supply. Therefore, the DIN regime using N1 fertilization was suggested to be adopted as the appropriate N-fertigation strategy for enhancing plant WUE responding to the resource-limited and CO_2_-elevated environment.

## Figures and Tables

**Figure 1 plants-13-02373-f001:**
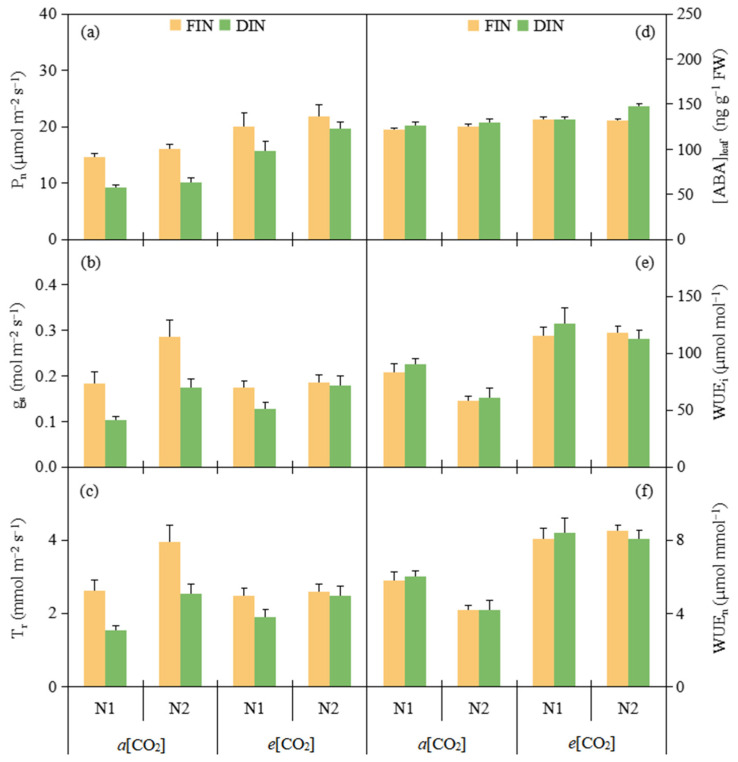
Leaf photosynthetic rate (P_n_) (**a**), stomatal conductance (g_s_) (**b**), transpiration rate (T_r_) (**c**), abscisic acid ([ABA]_leaf_) (**d**), intrinsic water use efficiency (WUE_i_) (**e**) and instantaneous water use efficiency (WUE_n_) (**f**) of tomato plants were affected by [CO_2_] concentration (*a*[CO_2_] and *e*[CO_2_]), N-fertigation regime (full irrigation during N-fertigation, FIN; deficit irrigation during N-fertigation, DIN) and N fertilizer level (N1 and N2). Error bars indicate standard error of the mean (n = 4). The results of ANOVA are shown in Table 1.

**Figure 2 plants-13-02373-f002:**
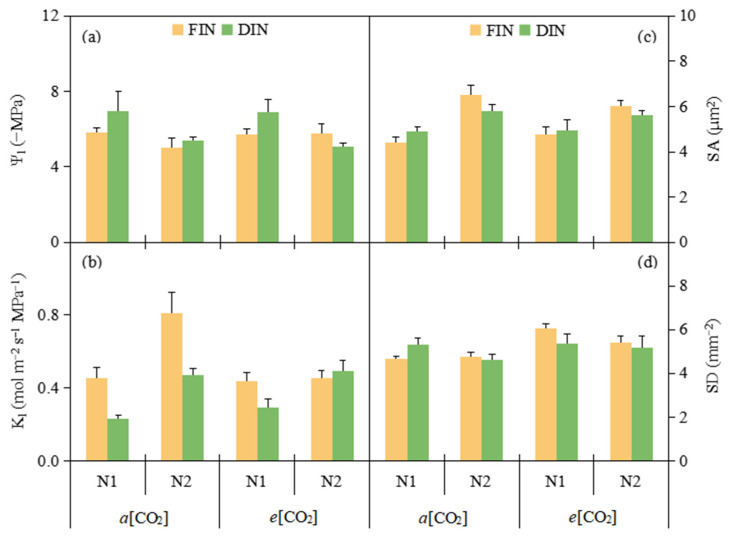
Leaf water potential (Ψ_l_) (**a**), leaf hydraulic conductivity (K_l_) (**b**), stomatal aperture (SA) (**c**) and stomatal density (SD) (**d**) of tomato plants were affected by [CO_2_] concentration (*a*[CO_2_] and *e*[CO_2_]), N-fertigation regime (full irrigation during N-fertigation, FIN; deficit irrigation during N-fertigation, DIN) and N fertilizer level (N1 and N2). Error bars indicate standard error of the mean (n = 4). The results of ANOVA are shown in Table 1.

**Figure 3 plants-13-02373-f003:**
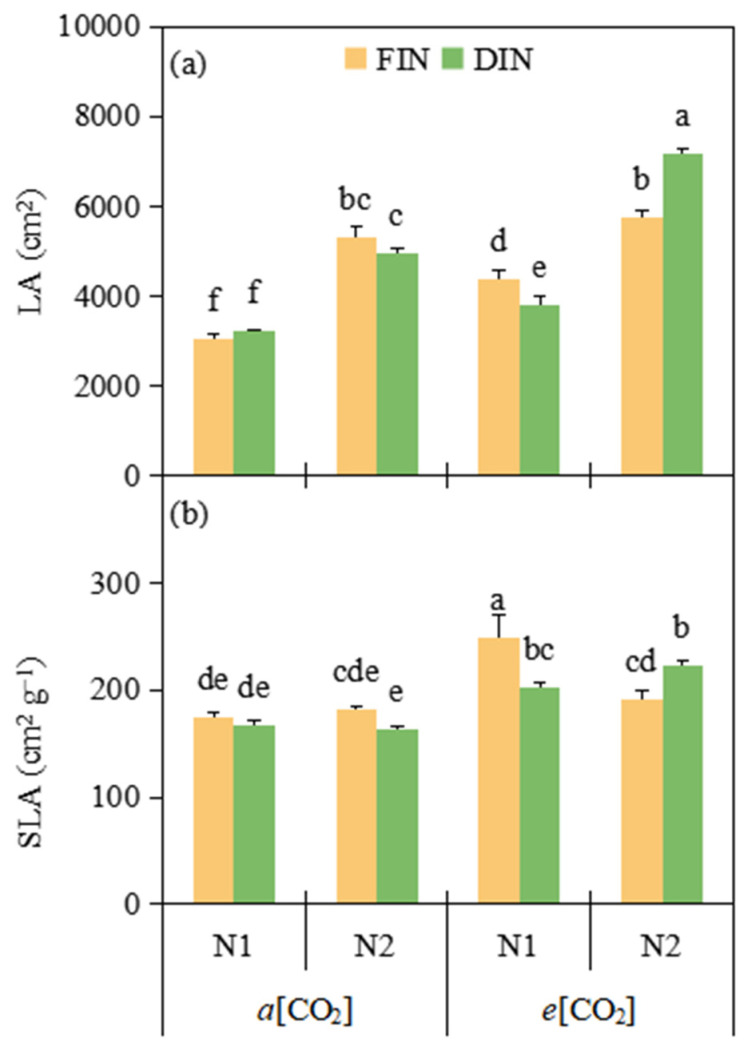
Leaf area (LA) (**a**) and specific leaf area (SLA) (**b**) of tomato plants were affected by [CO_2_] concentration (*a*[CO_2_] and *e*[CO_2_]), N-fertigation regime (full irrigation during N-fertigation, FIN; deficit irrigation during N-fertigation, DIN) and N fertilizer level (N1 and N2). Error bars indicate standard error of the mean (n = 4). The results of ANOVA are shown in Table 3. Different letters after the means indicate significant differences among treatments determined by Tukey’s multiple range test at *p* < 0.05.

**Figure 4 plants-13-02373-f004:**
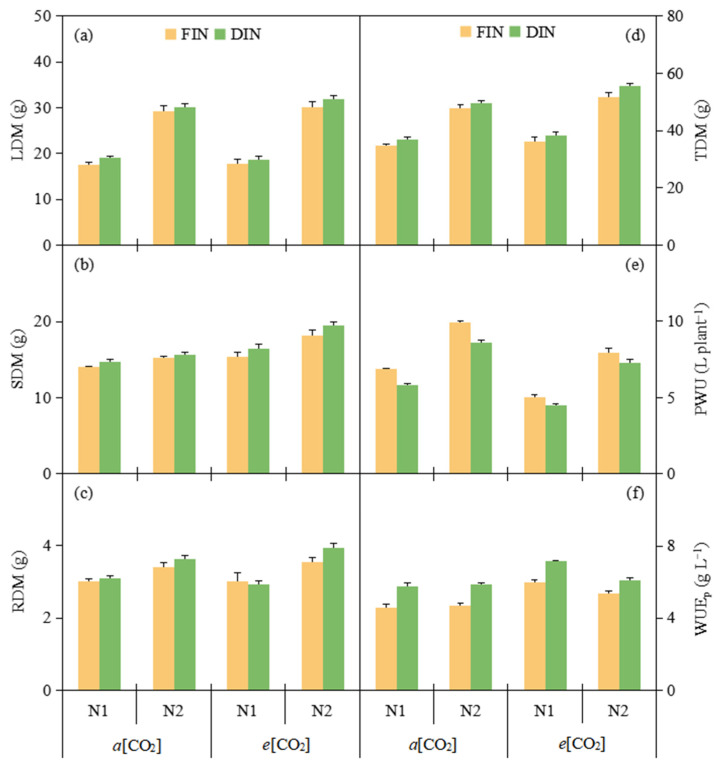
Leaf dry matter (LDM) (**a**), stem dry matter (SDM) (**b**), root dry matter (RDM) (**c**), total dry matter (TDM) (**d**), plant water use (PWU) (**e**) and plant water use efficiency (WUEp) (**f**) of tomato plants were affected by [CO_2_] concentration (*a*[CO_2_] and *e*[CO_2_]), N-fertigation regime (full irrigation during N-fertigation, FIN; deficit irrigation during N-fertigation, DIN) and N fertilizer level (N1 and N2). Error bars indicate standard error of the mean (n = 4). The results of ANOVA are shown in Table 3.

**Figure 5 plants-13-02373-f005:**
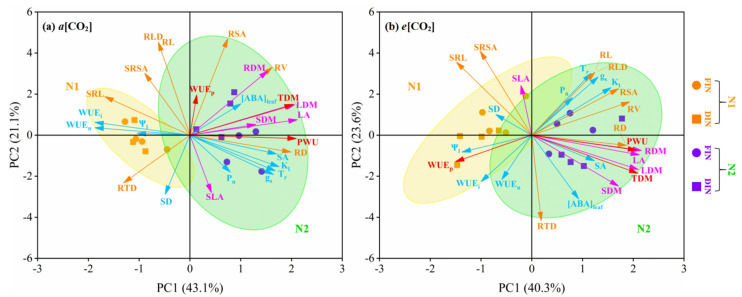
PCA diagram of tomato plants was affected by [CO_2_] concentration (*a*[CO_2_] and *e*[CO_2_]), N-fertigation regime (full irrigation during N-fertigation, FIN; deficit irrigation during N-fertigation, DIN) and N fertilizer level (N1 and N2). Ultramarine vectors were related to P_n_, g_s_, T_r_, [ABA]_leaf_, WUE_i_, WUE_n_, Ψ_l_, K_l_, SA and SD; amaranth vectors were related to LA, SLA, LDM, SDM and RDM; aurantia vectors were related to RL, RD, RSA, RV, SRL, SRSA, RLD and RTD; red vectors were related to TDM, PWU and WUE_p_.

**Table 1 plants-13-02373-t001:** Results of three-way ANOVA for leaf photosynthetic rate (P_n_), stomatal conductance (g_s_), transpiration rate (T_r_), abscisic acid ([ABA]_leaf_), intrinsic water use efficiency (WUE_i_), instantaneous water use efficiency (WUE_n_), leaf water potential (Ψ_l_), leaf hydraulic conductivity (K_l_), stomatal aperture (SA) and stomatal density (SD) of tomato plants were affected by [CO_2_] concentration (*a*[CO_2_] and *e*[CO_2_]), N-fertigation regime (full irrigation during N-fertigation, FIN; deficit irrigation during N-fertigation, DIN) and N fertilizer level (N1 and N2). The detailed data are shown in Figure 1 and Figure 2.

Factors	P_n_	g_s_	T_r_	[ABA]_leaf_	WUE_i_	WUE_n_	Ψ_l_	K_l_	SA	SD
[CO_2_]	***	ns	ns	***	***	***	ns	ns	ns	**
[FN]	***	***	***	*	ns	ns	ns	***	ns	ns
[N]	*	**	**	*	**	*	*	***	***	ns
[CO_2_] × [FN]	ns	*	*	ns	ns	ns	ns	*	ns	ns
[CO_2_] × [N]	ns	ns	*	ns	ns	*	ns	*	ns	ns
[FN] × [N]	ns	ns	ns	ns	ns	ns	ns	ns	ns	ns
[CO_2_] × [FN] × [N]	ns	ns	ns	ns	ns	ns	ns	ns	ns	ns

Notes: the *, ** and *** indicate significance levels at *p* < 0.05, *p* < 0.01 and *p* < 0.001, ns denoted no significance.

**Table 2 plants-13-02373-t002:** Root length (RL), root diameter (RD), root surface area (RSA), root volume (RV), specific root length (SRL), specific root surface area (SRSA), root length density (RLD) and root tissue density (RTD) of tomato plants were affected by [CO_2_] concentration (*a*[CO_2_] and *e*[CO_2_]), N-fertigation regime (full irrigation during N-fertigation, FIN; deficit irrigation during N-fertigation, DIN) and N fertilizer level (N1 and N2). The three-way ANOVA was used to analyze the significant probabilities for variables and the interaction of factors.

[CO_2_]	[FN]	[N]	RL (m)	RD (µm)	RSA (cm^2^)	RV (cm^3^)	SRL (m g^−1^)	SRSA (cm^2^ g^−1^)	RLD (cm cm^−3^)	RTD (mg cm^−3^)
*a*[CO_2_]	FIN	N1	223.53 ± 8.10	310.56 ± 5.12	2149.66 ± 57.74	16.56 ± 0.33	74.04 ± 2.52 a	711.98 ± 15.64 a	4.97 ± 0.18	182.40 ± 1.98
		N2	191.88 ± 14.66	371.53 ± 13.45	2169.06 ± 127.97	19.70 ± 0.84	56.21 ± 3.18 d	636.01 ± 23.28 b	4.26 ± 0.33	173.19 ± 4.12
	DIN	N1	211.36 ± 9.49	319.42 ± 4.67	2089.22 ± 72.19	16.53 ± 0.42	68.36 ± 3.37 abc	675.92 ± 28.73 ab	4.70 ± 0.21	187.80 ± 7.46
		N2	223.91 ± 10.31	355.86 ± 6.19	2434.21 ± 138.69	21.29 ± 1.52	61.79 ± 1.62 cd	670.94 ± 20.93 ab	4.98 ± 0.23	171.62 ± 7.58
*e*[CO_2_]	FIN	N1	219.20 ± 15.75	312.70 ± 4.29	2101.56 ± 176.43	16.17 ± 1.55	72.68 ± 1.92 ab	695.18 ± 21.19 ab	4.87 ± 0.35	188.24 ± 7.23
		N2	231.50 ± 11.10	326.29 ± 4.52	2332.94 ± 118.96	18.90 ± 1.12	65.09 ± 2.35 bc	655.54 ± 22.41 ab	5.14 ± 0.25	189.28 ± 7.39
	DIN	N1	206.29 ± 7.89	305.95 ± 2.52	1946.95 ± 76.25	14.70 ± 0.61	70.22 ± 1.65 ab	662.68 ± 15.28 ab	4.58 ± 0.18	200.28 ± 5.27
		N2	217.64 ± 14.69	331.67 ± 5.85	2240.50 ± 140.74	18.47 ± 1.15	55.08 ± 2.49 d	566.91 ± 20.46 c	4.84 ± 0.33	214.68 ± 6.99
ANOVA factor										
[CO_2_]			ns	***	ns	ns	ns	ns	ns	***
[FN]			ns	ns	ns	ns	ns	ns	ns	*
[N]			ns	***	*	***	***	**	ns	ns
[CO_2_] × [FN]			ns	ns	ns	ns	ns	ns	ns	ns
[CO_2_] × [N]			ns	**	ns	ns	ns	ns	ns	*
[FN] × [N]			ns	ns	ns	ns	ns	ns	ns	ns
[CO_2_] × [FN] × [N]			ns	ns	ns	ns	*	*	ns	ns

Notes: the *, ** and *** indicate significance levels at *p* < 0.05, *p* < 0.01 and *p* < 0.001, ns denoted no significance. Different letters after the means indicate significant differences among treatments determined by Tukey’s multiple range test at *p* < 0.05.

**Table 3 plants-13-02373-t003:** Results of three-way ANOVA for leaf area (LA), specific leaf area (SLA), leaf dry matter (LDM), stem dry matter (SDM), root dry matter (RDM), total dry matter (TDM), plant water use (PWU) and plant water use efficiency (WUE_p_) of tomato plants were affected by [CO_2_] concentration (*a*[CO_2_] and *e*[CO_2_]), N-fertigation regime (full irrigation during N-fertigation, FIN; deficit irrigation during N-fertigation, DIN) and N fertilizer level (N1 and N2). The detailed data are shown in Figure 3 and Figure 4.

Factors	LA	SLA	LDM	SDM	RDM	TDM	PWU	WUE_p_
[CO_2_]	***	***	ns	***	ns	**	***	***
[FN]	ns	ns	*	*	ns	**	***	***
[N]	***	ns	***	***	***	***	***	**
[CO_2_] × [FN]	*	ns	ns	ns	ns	ns	*	ns
[CO_2_] × [N]	ns	ns	ns	*	ns	*	ns	***
[FN] × [N]	**	*	ns	ns	ns	ns	ns	ns
[CO_2_] × [FN] × [N]	***	**	ns	ns	ns	ns	ns	ns

Notes: the *, ** and *** indicate significance levels at *p* < 0.05, *p* < 0.01 and *p* < 0.001, ns denoted no significance.

## Data Availability

The data related to this study are shown in the paper.

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
