# Peer review of "Effect of CO2 Elevation on Tomato Gas Exchange, Root Morphology and Water Use Efficiency under Two N-Fertigation Levels"

_plants, 2024, doi:10.3390/plants13172373_

Round 1

Reviewer 1 Report

Comments and Suggestions for Authors

The manuscript ‘Effect of CO2 elevation on tomato gas exchange, root morphology and water use efficiency under two N-fertigation levels’ by Zhang et al., deals with the effects of elevated CO2 on plant gas exchange, root morphology and water use efficiency under two N-fertigation levels. On the basis of the results, authors concluded that e[CO2] coupled with DIN strategy induced the lower stomatal aperture, higher stomatal density of N1 plant, synergistically decreased stomatal conductance and transpiration rate, increased leaf ABA, leading to the improved water use efficiency at stomatal, leaf and plant scales using two N fertilizer levels, particularly under N1 supply. This topic was suitable for the journal readers and the authors gathered an interesting and informative dataset.

Some concerns could be under consideration in this manuscript for minor revising:

1) Abstract: Please include specific results with relevant data.

2) In the Keywords: no “I. Introduction”, please delete it.

3) The keyword order was better to be Elevated CO2; N-fertigation…

4) Line 107, suggest adding ‘and’ before ‘tomato plant WUE’.

5) For the “Results” section, please include more detailed information, such as specific data. Additionally, provide a comparative analysis, clearly presenting the data to highlight the differences and similarities.

6) Line 198, ‘(FN)’ could change to ‘([FN])’ to align with charts.

7) Line 308, change ‘FN’ to ‘[FN]’.

8) Line 316, change ‘In’ to ‘in’.

9) Replace ‘date’ to ‘data’ in the last sentence of heading in Table 1 and Table 2.

10) Figure 6 heading, replace ‘plant’ to ‘plants’ in the first sentence to consist with the above Figure headings. Add the comma between WUEn and Ψl in the second sentence.

Comments on the Quality of English Language

It is good

Author Response

1) Abstract: Please include specific results with relevant data.

Thanks for the suggestion.

Yes, specific results with relevant data have been presented in the Abstract of the revised manuscript.

2) In the Keywords: no “I. Introduction”, please delete it.

3) The keyword order was better to be Elevated CO2; N-fertigation…

4) Line 107, suggest adding ‘and’ before ‘tomato plant WUE’.

Thanks for the suggestions.

2) Yes, these words have been deleted in the Keywords.

3) Yes, the keyword order has been changed.

4) Line 107: Yes, ‘and’ has been added before ‘tomato plant WUE’.

5) For the “Results” section, please include more detailed information, such as specific data. Additionally, provide a comparative analysis, clearly presenting the data to highlight the differences and similarities.

Thanks for the suggestion.

Yes, the specific data and comparative data has been clearly presented in the Results section, the details have been modified in the revised manuscript.

6) Line 198, ‘(FN)’ could change to ‘([FN])’ to align with charts.

7) Line 308, change ‘FN’ to ‘[FN]’.

8) Line 316, change ‘In’ to ‘in’.

9) Replace ‘date’ to ‘data’ in the last sentence of heading in Table 1 and Table 2.

10) Figure 6 heading, replace ‘plant’ to ‘plants’ in the first sentence to consist with the above Figure headings. Add the comma between WUEn and Ψl in the second sentence.

Thanks for the suggestions.

6), 7), 8), 9) Line 198, Line 308, Line 316: Yes, these words have been changed into correct words.

10) Yes, in Figure 6 heading, the word has been changed into correct word and the comma between WUEn and Ψl in the second sentence has been added in the revised manuscript.

Reviewer 2 Report

Comments and Suggestions for Authors

Lines 16-20. The past tense gave the impression that the author was reporting the results of the current manuscript. Change the verb tense to the present; instead of “reduced”, use “reduces”. Thus, rewrite the verb tenses for the beginning of the abstract.

Lines 31-34. Here, also delete "some". Change "These findings would provide some valuable acknowledgment for optimizing water and N fertilizer management and improving plant water use efficiency responding to the potential resource-limited and CO2-enriched scenario." to " These findings provide valuable information to optimize water and nitrogen fertilizer management and improve plant water use efficiency, responding to the potential resource-limited and CO2-enriched scenario."

Overall, the introduction is good. But, insert a paragraph about the importance of tomatoes at the global level.

Table and figure captions should be self-explanatory. Therefore, I suggest that the authors clearly describe the treatments and then the abbreviation: ambient CO2 concentration (a[CO2]) and elevated CO2 concentration (e[CO2]).

Based on the analysis of variance, perform the Tukey test when there was significance (P ≤ 0.05) in the interaction [CO2]×[FN]×[N] and show the letters to indicate the best answer. E.g., do this for SRL, SRSA, LA, and SLA.

In my opinion, correlation analysis does not make sense in the work, since PCA (Principal Component Analysis) is a multivariate analysis technique that can be used to analyze interrelationships between a large number of variables and explain these variables in terms of their inherent dimensions (Components).

Author Response

Lines 16-20. The past tense gave the impression that the author was reporting the results of the current manuscript. Change the verb tense to the present; instead of “reduced”, use “reduces”. Thus, rewrite the verb tenses for the beginning of the abstract.

Thanks for the suggestion.

Lines 16-20: Yes, the verb tenses for the beginning of the Abstract has been changed in the revised manuscript.

Lines 31-34. Here, also delete "some". Change "These findings would provide some valuable acknowledgment for optimizing water and N fertilizer management and improving plant water use efficiency responding to the potential resource-limited and CO2-enriched scenario." to " These findings provide valuable information to optimize water and nitrogen fertilizer management and improve plant water use efficiency, responding to the potential resource-limited and CO2-enriched scenario."

Thanks for the suggestion.

Lines 31-34: Yes, the word has been deleted and the sentence has been changed in the revised manuscript.

Overall, the introduction is good. But, insert a paragraph about the importance of tomatoes at the global level.

Thanks for the suggestion.

Yes, the content about the importance of tomato at the global level has been presented at the beginning of the last paragraph in Introduction section.

Table and figure captions should be self-explanatory. Therefore, I suggest that the authors clearly describe the treatments and then the abbreviation: ambient CO2 concentration (a[CO2]) and elevated CO2 concentration (e[CO2]).

Thanks for the suggestion.

Yes, the Table and Figure captions have been changed to ambient CO2 concentration (a[CO2]) and elevated CO2 concentration (e[CO2]) in the revised manuscript.

Based on the analysis of variance, perform the Tukey test when there was significance (P ≤ 0.05) in the interaction [CO2]×[FN]×[N] and show the letters to indicate the best answer. E.g., do this for SRL, SRSA, LA, and SLA.

Thanks for the suggestion.

Yes, the Tukey test of interaction [CO2]×[FN]×[N] on SRL, SRSA, LA and SLA have been performed and the letters indicated the best answer have been revealed in Table 3 and Figure 3 of the revised manuscript.

In my opinion, correlation analysis does not make sense in the work, since PCA (Principal Component Analysis) is a multivariate analysis technique that can be used to analyze interrelationships between a large number of variables and explain these variables in terms of their inherent dimensions (Components).

Thanks for the suggestion.

Yes, the relevant content of correlation analysis has been removed in the revised manuscript.
